# Enhanced Intestinal Permeability and Plasma Concentration of Metformin in Rats by the Repeated Administration of Red Ginseng Extract

**DOI:** 10.3390/pharmaceutics11040189

**Published:** 2019-04-18

**Authors:** Sojeong Jin, Sowon Lee, Ji-Hyeon Jeon, Hyuna Kim, Min-Koo Choi, Im-Sook Song

**Affiliations:** 1College of Pharmacy, Dankook University, Cheon-an 31116, Korea; astraea327@naver.com; 2College of Pharmacy and Research Institute of Pharmaceutical Sciences, Kyungpook National University, Daegu 41566, Korea; okjin917@hanmail.net (S.L.); kei7016@naver.com (J.-H.J.); hyuna6672@gmail.com (H.K.)

**Keywords:** red ginseng extract (RGE), metformin, organic cation transporter, herb–drug interaction, intestinal permeability

## Abstract

We aimed to assess the potential herb–drug interactions between Korean red ginseng extract (RGE) and metformin in rats in terms of the modulation of metformin transporters, such as organic cation transporter (Oct), multiple toxin and extrusion protein (Mate), and plasma membrane monoamine transporter (Pmat). Single treatment of RGE did not inhibit the in vitro transport activity of OCT1/2 up to 500 µg/mL and inhibited MATE1/2-K with high IC_50_ value (more than 147.8 µg/mL), suggesting that concomitant used of RGE did not directly inhibit OCT- and MATE-mediated metformin uptake. However, 1-week repeated administration of RGE (1.5 g/kg/day) (1WRA) to rats showed different alterations in mRNA levels of Oct1 depending on the tissue type. RGE increased intestinal Oct1 but decreased hepatic Oct1. However, neither renal Oct1/Oct2 nor Mate1/Pmat expression in duodenum, jejunum, ileum, liver, and kidney were changed in 1WRA rats. RGE repeated dose also increased the intestinal permeability of metformin; however, the permeability of 3-*O*-methyl-d-glucose and Lucifer yellow was not changed in 1WRA rats, suggesting that the increased permeability of metformin by multiple doses of RGE is substrate-specific. On pharmacokinetic analysis, plasma metformin concentrations following intravenous injection were not changed in 1WRA, consistent with no significant change in renal Oct1, Oct2, and mate1. Repeated doses of RGE for 1 week significantly increased the plasma concentration of metformin, with increased half-life and urinary excretion of metformin following oral administration of metformin (50 mg/kg), which could be attributed to the increased absorption of metformin. In conclusion, repeated administration of RGE showed in vivo pharmacokinetic herb–drug interaction with metformin, with regard to its plasma exposure and increased absorption in rats. These results were consistent with increased intestinal Oct1 and its functional consequence, therefore, the combined therapeutic efficacy needs further evaluation before the combination and repeated administration of RGE and metformin, an Oct1 substrate drug.

## 1. Introduction

Diabetes mellitus is a major health-threatening disease because the number of newly diagnosed diabetic patients is sharply increasing worldwide [1]. The duration and level of hyperglycemia in type 2 diabetic patients are highly associated with the risk of diabetic complications [2]. Therefore, achieving glycemic control in the early stages of therapy is very important for diabetic complications in type 2 diabetes mellitus. For this, using add-on therapy with other drugs that have a different mode of action alongside metformin has been recommend and metformin is the most prescribed anti-diabetic, as either mono or combination therapy with other drugs in type 2 diabetic patients [3].

Metformin is extensively eliminated via renal route as a parent form, in which renal tubular secretion mediated by cationic transporters is involved [4]. Among cationic transporters, organic cation transporter 2 (OCT2 in humans; Oct2 in rats) and multiple toxin and extrusion protein 1/2 (MATE1/2 in humans and Mate1/2 in rats) were responsible for the active renal excretion of metformin [5]. OCT1/Oct1 mediated the hepatic and intestinal distribution of metformin, and this is crucial for the phosphorylation of adenosine monophosphate (AMP)-activated protein kinase (AMPK), an enzyme critical for the regulation of glucose and lipid metabolism, energy imbalance [3,6,7,8,9], and the regulation of gluconeogenesis [7]. In the metformin absorption process, cationic transporters were also involved. Approximately 20% of the metformin dose was absorbed in the duodenum and 60% was absorbed in the jejunum and ileum in a dose-dependent and -independent manner [5]. In the dose-dependent process, OCT1/Oct1, MATE1/Mate1, and plasma membrane monoamine transporter (PMAT in humans; Pmat in rats) are involved in a coordinated manner [10]. 

Therefore, the importance of cationic transporters, such as Oct, Mate, and Pmat, in the efficacy and pharmacokinetics (PK) of metformin suggests that it is reasonable to expect herb–drug interactions of metformin with concomitantly administered Oct modulators [11].

Korean red ginseng may benefit mice with streptozotocin-induced diabetes owing to its hypoglycemic and immunomodulatory effects [12]. Supplementation with Korean red ginseng extract (RGE), fermented red ginseng, and black ginseng also has been shown to improve type 1 and type 2 diabetic conditions in both animals [12,13,14] and humans [15,16,17]. 

The underlying mechanism of action of red or black ginseng is probably mediated via the AMPK pathway [18,19], reduction of reactive oxygen species, and anti-inflammatory cytokines [12]. Black ginseng also inhibited inflammatory responses and induced beta-cell apoptosis in mice with streptozotocin-induced diabetes [13]. Red ginseng increased the expression of insulin receptors, and thereby increased insulin sensitivity [20]. Ginseng root treatment increased glucose uptake into 3T3-L1 adipocytes or skeletal muscle cells via the upregulation of glucose transporter 4 (GLUT4) through the activation of the peroxisome proliferator agonist receptor (PPAR)-γ [21,22]. These multiple anti-diabetic mechanisms of ginseng are due to the anti-diabetic mechanisms of various ginsenosides, the pharmacologically active components of ginseng or ginseng products [21,23]. 

We previously reported that the repeated administration of 1.5 g/kg to rats decreased Mrp2 expression and altered the elimination of methotrexate, a substrate drug of Mrp2. The concentrations of ginsenoside Rb1, Rb2, Rc, and Rd in rat plasma were comparable after 1 and 2 weeks of RGE multiple treatment (1.5 g/kg) [24] and the mRNA levels of efflux transporters, such as P-gp, Mrps, and Bcrp, were also comparable after 1 and 2 weeks of RGE multiple treatment (1.5 g/kg). Based on the previous report, we selected the dose and treatment period of RGE as repeated oral administration of RGE 1.5 g/kg for 1 week. The selected RGE dose in this study is in the range that shows anti-diabetic effect. In numerous animal studies that showed anti-diabetic effect of RGE, the RGE dose has ranged from 200 mg/kg to 2.0 g/kg (i.e., 3–15 mg/kg of total ginsenosides) [14,25,26,27]. In human studies, RGE was administered to diabetic patients for 4 to 24 weeks at doses of 2.7 g–6.0g/day, which usually contained 50–100 mg ginsenosides/day [16,17,23]. 

Therefore, in this study, we investigated the effect of RGE (1.5 g/kg) on the transport activities and expression of these cationic transporters as a crucial modulator for RGE-mediated herb–drug interaction on metformin PK. To further investigate the in vivo herb–drug interaction potential, the effect of single or multiple administrations of RGE (1.5 g/kg) for 1 week on the PK of metformin was assessed.

## 2. Materials and Methods

### 2.1. Materials

RGE was obtained from Punggi Ginseng Cooperative Association (Punggi, Korea). RGE contains >60% of dried ginseng. This product was produced under the current guidelines of the Korea Good Manufacturing Practice. 

Metformin, cimetidine, 3-*O*-methyl-d-glucose, and Lucifer yellow were purchased from Sigma–Aldrich Chemical Co. (St. Louis, MO, USA). The [^14^C]metformin (110 mCi/mmol) was purchased from Moravek (Brea, CA, USA). The 3-*O*-[^14^C-methyl]-d-glucose (40 mCi/mmol) was purchased from Perkin Elmer Inc. (Boston, MA, USA). All other chemicals and solvents were reagent or analytical grade.

### 2.2. Inhibitory effects of Red Ginseng Extract (RGE) on the Organic Cation Transport Activities 

HEK293 cells overexpressing OCT1, OCT2, MATE1, and MATE2-K (HEK293-OCT1, -OCT2, MATE1, and MATE2-K, respectively; Corning, Tewksbury, MA, USA) were used and characterized as previously described [11,28]. Cells were maintained in Dulbecco’s modified Eagle’s medium (DMEM) supplemented with 10% fetal bovine serum, 5 mM nonessential amino acid, and 100 U/mL penicillin-streptomycin at 37 °C in 8% CO_2_ condition. HEK293- OCT1, OCT2, MATE1, or MATE2-K cells were seeded at a density of 1 × 10^5^ cells/well in poly-d-lysine coated 24‑well plates. In HEK293-MATE1 or -MATE2-K cells, 2 mM sodium butyrate was also supplemented in the culture medium. After 24 h, the growth medium was discarded from the cells and the cells were washed with pre-warmed Hank’s balanced salt solution (HBSS; Sigma, St. Louis, MO, USA) and incubated for 30 min in HBSS. In HEK293-MATE1, or -MATE2-K cells, 40 mM ammonium chloride was also added for the outwardly directed proton gradient. After replacing incubation buffer with fresh HBSS containing 10 µM [^14^C]metformin with or without cimetidine (0.01 µM–10 mM) or RGE (0.1–500 µg/mL), the uptake of [^14^C]metformin into the HEK293 cells overexpressing OCT1, OCT2, MATE1, or MATE2-K was measured for 5 min. After the 5 min incubation, cells were washed three times with 200 µL of ice-cold HBSS immediately after placing the plates on ice. Then, the cells were lysed with 10% sodium dodecyl sulfate (SDS) and the cell lysates were mixed with Optiphase cocktail solution. Thereafter, the radioactivity of the cell lysates was measured using a liquid scintillation counter (Microbeta 2, Perkin Elmer Inc., Boston, MA, USA).

### 2.3. Animals and Ethical Approval

Male Sprague-Dawley rats (7–8-weeks-year-old, 220–250 g) were purchased from Samtako Co. (Osan, Korea). Animals were acclimatized for 1 week in an animal facility at Kyungpook National University. Food and water were available ad libitum. All animal procedures were approved by the Animal Care and Use Committee of Kyungpook National University (Approval No. 2017-0021) and carried out in accordance with the National Institutes of Health guidance for the care and the use of laboratory animals. 

Rats were randomly divided into the control group (*n* = 16), single RGE administration group (SA, *n* = 16), and repeated RGE administration for 1 week group (1WRA, *n* = 16). The control group received water (2 mL/kg) at 09:00 for 7 days using oral gavage. SA group received water (2 mL/kg) for 6 days and received RGE suspension (1.5 g/kg/day as 2 mL/kg suspended in water) on seventh day orally at 09:00. Rats from 1WRA group received RGE suspension (1.5 g/kg/day as 2 mL/kg suspended in water) at 09:00 for 7 days via oral gavage. The rats were fasted for at least 12 h before the dissection of duodenal, jejunal, and ileal segments or oral administration of metformin, but had free access to water. 

### 2.4. Analysis of Ginsenosides in RGE Extracts

Ginsenoside concentrations in RGE and plasma samples were analyzed using an Agilent 6470 triple quadrupole liquid chromatography–tandem mass spectrometry (LC–MS/MS) system (Agilent, Wilmington, DE, USA) equipped with an Agilent 1260 high performance liquid chromatography (HPLC) system, according to a previously published method [24]. Briefly, diluted RGE and plasma samples (50 μL) were vigorously mixed with 200 μL methanol containing berberine (0.5 ng/mL; used as an internal standard) and centrifuged at 10,000× *g* for 10 min at 4 °C. Aliquots (20 μL) of the supernatant were injected into the LC–MS/MS system for the analysis of ginsenosides. Ginsenosides were separated on an Omega Polar C18 column (2.1 × 100 mm, 3 μm particle size; Phenomenex, Torrence, CA, USA) using a mobile phase consisting of water (A) and methanol (B), containing 0.1% formic acid at a flow rate of 0.22 mL/min. The solvent gradient program was as follows: (1) 0–0.5 min, 72% B; (2) 0.5–8.5 min, 84% B; (3) 8.5–19.0 min, 72%.

Quantification of a separated ginsenoside peak was performed at *m*/*z* 1131.6 → 365.1 for Rb1 (*T*_R_ (retention time) 5.0 min), *m*/*z* 1101.6 → 335.1 for Rb2 (*T*_R_ 6.4 min) and Rc (*T*_R_ 5.2 min), *m*/*z* 969.9 → 789.5 for Rd (*T*_R_ 8.2 min) and Re (*T*_R_ 1.6 min), *m*/*z* 824 → 643.6 for Rg1 (*T*_R_ = 1.7 min), *m*/*z* 807.5 → 365.1 for Rg3 (*T*_R_ 12.3 min), *m*/*z* 661.5 → 203.1 for Rh1 (*T*_R_ 3.2 min) and F1 (*T*_R_ 3.9 min), *m*/*z* 587.4 → 407.4 for Rh2 (*T*_R_ 11.5 min), *m*/*z* 807.5 → 627.5 for F2 (*T*_R_ 5.3 min), *m*/*z* 645.5 → 203.1 for compound K (*T*_R_ 9.5 min), *m*/*z* 425.4 → 109.1 for protopanaxadiol (*T*_R_ 18.8 min), *m*/*z* 441.3 → 109.1 for protopanaxatriol (*T*_R_ 2.9 min), and *m*/*z* 336.1 → 320 for berberine (internal standard, IS) (*T*_R_ 3.7 min) in the positive ionization mode with collision energy (CE) of 15–65 eV. The plasma and extract calibration standards of 14 ginsenosides were 1–100 ng/mL and intraday and interday precision and accuracy were less than 12.1% in all samples.

### 2.5. Effect of RGE on mRNA and Protein Expression of Oct1 and Oct2 in the Intestine, Liver, and Kidney

Relative mRNA expression of Oct1, Oct2, Mate1, and Pmat in the duodenum, jejunum, ileum, liver, and kidney from rats of control, SA, and 1WRA groups was measured using quantitative real-time reverse-transcription polymerase chain reaction (qRT-PCR) analysis. Two hours after the single RGE dose and 24 h after the last RGE dose, the duodenum was excised about 10 cm and jejunum was excised about 10 cm after the duodenum section. The ileal segment (about 20 cm) was excised following jejunal dissection after the rats were euthanized by cervical dislocation. The dissected duodenal, jejunal, and ileal segments were washed using a 10 mL syringe filled with pre-warmed saline (30 mL). Enterocytes from duodenum, jejunum, and ileum segments were scraped up using slide glasses. The liver and kidney were also isolated after a gentle wash with pre-warmed saline, minced, and snap-frozen in liquid nitrogen. Enterocytes from the duodenum, jejunum, and ileum, and tissue samples from the liver and kidney (approximately 100 mg) were homogenized with 1.0 mL TRIzol (Invitrogen, Carlsbad, CA, USA) using a Wheaton Tissue Grinder (DWK Life Sciences, Millville, NJ, USA). Total RNA was extracted using 100 μL bromochloropropane followed by vigorous mixing for 10 s, incubating for 15 min on ice, and centrifuged at 10,000× *g* for 10 min at 4 °C. RNA in aqueous phase was precipitated with 100 μL isopropanol and washed twice with 75% ethanol. The concentration of total RNA was determined by Nano Vue Plus (GE healthcare Korea, Seoul, Korea). The quantitated RNA was diluted with sterile diethyl pyrocarbonate-treated water and stored at −80 °C.

The qRT-PCR was conducted with 10 ng of RNA using a TaqMan RNA-to-CT 1-Step Kit and TaqMan Gene Expression Assays Kit (ThermoFisher scientific Korea, Seoul, Korea). RT step was performed at 48 °C for 25 min. Each PCR program started with enzyme activation at 95 °C for 10 min, followed by 40 cycles of denaturation at 95 °C for 15 s and annealing and extension at 60 °C for 60 s. Predesigned TaqMan Gene expression assay for rat Oct1 (slc22a1, NM_012697.1, Rn00562250_m1), Oct2 (slc22a2, NM_031584.1, Rn00580893_m1), Mate1 (slc47a1, NM_001014118.2, Rn01460731_m1), Pmat (slc29a4, NM_001105911.1, Rn01453824_m1), and β-actin (NM_031144.2, Rn00667869_m1) were used. The relative quantitation of the mRNA levels of Oct1, Oct2, Mate1, and Pmat was obtained as the threshold cycle (C_T_) values of the gene of interest and was normalized to the relative quantity of beta-actin, an endogenous internal standard ((ΔΔC_T_). The mRNA abundance of Oct1, Oct2, Mate1, and Pmat was calculated as 2^−(ΔΔCT)^ [29,30]. 

Tissue samples from the ileum, liver, and kidney (approximately 100 mg) were homogenized with 1.0 mL CETi lysis buffer containing protease inhibitors (4 mM AEBSF, 1 μg/mL benzamidine, 1 μg/mL leupeptin, 1 μg/mL pepstatin, 1 mM EDTA, 1 mM EGTA), and phosphatase inhibitors (1 mM sodium fluoride, 1 mM sodium orthovanadate, 1 mM β-glycerophosphate, 2.5 mM sodium pyrophosphate) (Translab, Daejeon, Korea) using a Wheaton Tissue Grinder and incubated for 20 min at 25 °C. Total protein was obtained by centrifugation at 10,000× *g* for 10 min at 4 °C. Protein concentration was measured using a Perce BCA protein assay kit (ThermoFisher Scientific Korea, Seoul, Korea) according to the manufacturer’s procedure. 

Aliquots containing 50 µg proteins were separated using SDS-polyacrylamide gel electrophoresis (PAGE) on a 4–15% gradient gel (Bio-Rad, Hercules, CA, USA) and then transferred onto a nitrocellulose membrane (Bio-Rad). The membrane was blocked with 5% bovine serum albumin in Tris-buffered saline containing 0.1% Tween 20 (TBST) for 1 h, followed by overnight incubation with primary anti-Oct1 (1:100, Santa Cruz Biotechnology, Dallas, TX, USA), anti-Oct2 (1:100, Santa Cruz Biotechnology), and β-actin antibody (1:1000, Santa Cruz Biotechnology) at 4 °C. The membrane was rinsed twice with TBST at 25 °C and treated with horseradish peroxidase (HRP)-labeled secondary antibody (1:1000, Santa Cruz Biotechnology) for 1 h. The membrane was rinsed three times with TBST at 25 °C and treated with Luminata Forte Western HRP substrate (Merck Millipore, Darmstadt, Germany) for 1 min. Western bands were visualized using ImageQuant LAS 4000 Mini (GE Healthcare Korea, Seoul, Korea).

### 2.6. Effect of RGE Treatment on the Permeability of Metformin 

The permeability of metformin with or without RGE treatment was measured according to the method of Kwon et al. [31]. Briefly, duodenum, jejunum, and ileum segments were dissected and washed as described earlier. The duodenal, jejunal, and ileal segments mounted on the inserts of the Ussing chambers (Navicyte, Holliston, MA, USA) were acclimatized with HBSS for 30 min. The experiments started with changing the HBSS with preheated 1 mL of HBSS containing 50 µM [^14^C]metformin to the apical side and fresh HBSS (1 mL) to the basal side. Aliquots (400 μL) of HBSS in the basal side were withdrawn at 0, 30, 60, 90, and 120 min, and the withdrawn samples were compensated with an equal volume of fresh, preheated HBSS. Carbogen gas (5% CO_2_/ 95% O_2_) was bubbled into the Ussing chambers at a rate of 150 drops/min during the experiment. Aliquots (50 μL) of the samples were mixed with Optiphase cocktail solution and the radioactivity of metformin in permeability samples was measured using a liquid scintillation counter. 

For the comparison, the permeability of 50 µM 3-*O*-[^14^C-methyl]-d-glucose, a substrate for glucose transporters [32], was also measured. For the generation of Na^+^ gradient, rat duodenal, jejunal, and ileal segments mounted on the inserts of the Ussing chamber were pretreated with Na^+^ free media comprising 10 mM HEPES, 5 mM Tris, 140 mM choline chloride, 2 mM KCl, 1 mM CaCl_2_, and 1 mM MgCl_2_ (pH7.4) for 30 min. After aspirating Na^+^ free media, the transport of 50 µM 3-*O*-[^14^C-methyl]-d-glucose was initiated by the addition of Na^+^ gradient buffer consisting of 10 mM HEPES, 5 mM Tris, 140 mM NaCl, 2 mM KCl, 1 mM CaCl_2_, and 1 mM MgCl_2_ (pH7.4) to the apical side. The following procedures were identical with metformin permeability. 

The effect of RGE on the paracellular permeability was measured using 50 µM Lucifer yellow, a marker compound for paracellular pathway [31] with the same protocol as metformin. A 100 μL aliquot of samples was transferred to 96-well plates and the fluorescence of Lucifer yellow was read directly in a fluorescence plate reader using a 485 nm excitation and an emission filter of 535 nm.

### 2.7. Effect of RGE on PK of Metformin in Rats

Rat also received metformin (0.25 mg/kg, 0.5 mL/kg in water) via tail vein 2 h after the single RGE treatment or 24 h after the last RGE or vehicle treatment. After the metformin injection, rats were kept in a metabolic cage for 24 h to collect the urine. Blood samples (approximately 100 µL) were collected at 0, 0.25, 0.5, 1, 1.5, 2, 4, and 8 h via the retro-orbital vein following metformin administration and centrifuged at 10,000× *g* for 10 min to separate the plasma. An aliquot (20 µL) of each plasma sample was stored at −80 °C for the metformin analysis. Urine samples were collected for 24 h in a metabolic cage and weighed. An aliquot (50 µL) of urine samples was stored at −80 °C for the analysis.

Rats received metformin (50 mg/kg, 2 mL/kg in water) via oral gavage 2 h after the single RGE treatment or 24 h after the last RGE or vehicle treatment. After the metformin administration, rats were kept in a metabolic cage for 24 h to collect the urine. Blood samples (approximately 100 µL) were collected at 0, 0.25, 0.5, 1, 2, 3, 4, 8, and 24 h via the retro-orbital vein following metformin administration and the samples were prepared and stored as described previously. 

Aliquots of plasma (20 µL) and urine (50 µL) samples were vigorously mixed with 200 μL methanol containing 20 ng/mL propranolol (internal standard, IS) for 10 min. After centrifugation at 10,000× *g* for 10 min, an aliquot (2 µL) was injected into the LC-MS/MS system.

### 2.8. LC-MS/MS Analysis of Metformin

The concentration of metformin was analyzed using a modified LC-MS/MS method, as previously reported by Kwon et al. [11]. An Agilent 6430 Triple Quad LC–MS/MS system coupled with an Agilent 1260 series high-performance liquid chromatography system was used. Samples were eluted through a Synergy Polar RP column (2.0 mm × 150 mm, 4 µm particle size, Phenomenex, Torrence, CA, USA) using a mobile phase that consisted of methanol and water (70:30, *v*/*v*) with 0.1% formic acid at a flow rate of 0.2 mL/min. Metformin and propranolol (IS) were detected at retention times of 2.09 min and 3.03 min, respectively, by an electrospray ionization with a positive ion mode. Quantification was carried out using selected reaction monitoring mode at *m*/*z* 130.2 → 71.4 for metformin, and *m*/*z* 260.0 → 116.0 for propranolol. Plasma and urine calibration standards for the measurement of metformin from oral administration were 0.1–15 μg/mL and intraday and interday precision and accuracy were less than 11.3% in all samples. For the measurement of metformin concentration from intravenous injection, the plasma and urine calibration standards were 5–2000 ng/mL and intraday and interday precision and accuracy were less than 10.8% in all samples. 

### 2.9. Data Analysis

In the inhibition studies, the percentages of the transport rate of metformin (*v*) with or without cimetidine or RGE were calculated and the data were fitted to an inhibitory effect model (i.e., $v=E_{\max}(1-\frac{\left[I\right]}{IC_{50}+[I]})$ [28]. *E*_max_ and [I] represent the percentage of transport rate of metformin without inhibitor and the concentration of inhibitor, respectively. The IC_50_ value indicated the concentration of the inhibitor showed half maximal inhibition.

Pharmacokinetic parameters were evaluated from plasma concentration versus time curves by using non-compartment analysis of WinNonlin (version 5.1; Pharsights, Cary, NC, USA). 

The apparent permeability (P_app_) was calculated by dividing the initial transport rate of metformin, 3-*O*-methyl-d-glucose, and Lucifer yellow by initial concentrations added to the mucosal side and the surface area of the insert [31].

The statistical significance was assessed by *t*-test using Statistical Package for the Social Sciences (version 24.0; SPSS Inc., Chicago, IL, USA). 

## 3. Results

### 3.1. Concentration of Ginsenosides in RGE and Rat Plasma

We previously developed analytical methods for the determination of 14 ginsenosides (Rb1, Rb2, Rc, Rd, Rh2, Rg3, F2, compound K, Protopanaxadiol, Re, Rh1, Rg1, F1, and Protopanaxatriol) using LC-MS/MS. The sum of ginsenosides was calculated as 8.3 mg/g RGE. The ginsenosides administered to rats were 12–13 mg/kg ginsenosides and in the range of previously administered ginsenoside contents for the treatment for streptozotocin-induced diabetic animals (3–15 mg/kg) [14,25,26,27].

Plasma concentrations of ginsenosides Rb1, Rb2, Rc, and Rd in rats after single or multiple administration of RGE (1.5 g/kg/day) for 1 week were similar as the previous results [24]. 

### 3.2. Inhibitory Effects of RGE on the Transport Activities of OCT1, OCT2, MATE1, and MATE2-K 

To investigate the herb–drug interaction potential between RGE and metformin, the inhibitory effect of RGE on the cationic transporter-mediated metformin uptake was measured. We firstly measured the uptake of 10 µM [^14^C]metformin in the presence of cimetidine, a representative inhibitor of OCT1, OCT2, MATE1, and MATE2-K [33,34].

Cimetidine inhibited the OCT1, OCT2, MATE1, and MATE2-K-mediated metformin uptake in a concentration-dependent manner and IC_50_ values of cimetidine were calculated as 61.8 μM, 197.3 μM, 0.55 μM, and 5.03 μM, respectively (Figure 1A–D), which were comparable with the IC_50_ values of cimetidine reported in the literature (i.e., 57 μM for OCT1; 110 μM for OCT2; 0.7 μM for MATE1; 4.8 μM for MATE2-K) [33,34].

Using the same experimental conditions, we measured the inhibitory effect of single treatment of RGE on the OCT1, OCT2, MATE1, and MATE2-K-mediated metformin uptake. Consequently, RGE did not inhibit the OCT1- and OCT2-mediated metformin uptake up to 500 µg/mL of RGE treatment (Figure 1E,F). However, RGE inhibited MATE1- and MATE2-K-mediated metformin uptake in a concentration-dependent manner and the IC_50_ values were calculated as 147.8 µg/mL and 181.9 µg/mL for MATE1 and MATE2-K, respectively (Figure 1G,H). 

Since the plasma concentration of ginsenosides after single or multiple treatment of RGE was much lower (in the range of 2.9–35.7 ng/mL, Table 1) than the IC_50_ values of RGE on the cationic transporter-mediated metformin uptake, the in vivo relevance of herb–drug interaction between RGE and metformin via the direct inhibition of cationic transport activities could be expected to be remote.

### 3.3. Effect of RGE on the Expression Level of Cationic Transporters

The effect of RGE treatment on the expressional modulation of cationic transporters in the tissues responsible for the metformin absorption, distribution, and elimination was investigated.

Repeated administration of RGE for 1 week increased the Oct1 mRNA in the duodenum, jejunum, and ileum but was significantly decreased in the liver without altering the Oct1 mRNA in the kidney (Figure 2A). Contrary to the case of Oct1, Oct2 mRNAs showed limited expression in duodenum, jejunum, ileum, and liver but showed highest expression in the kidney. But Oct2 mRNA levels in the kidney were not changed by the repeated administration of RGE (Figure 2B). The levels of Mate1, which is involved in the excretion of metformin into the intestinal lumen, bile, and urine, were not altered by the 1-week RGE treatment in all tissues (Figure 2C). The levels of Pmat, which showed the lowest expression levels among the cationic transporters tested in this study, were not altered by RGE treatment (Figure 2D). In all cases, mRNA expression levels of Oct1, Oct2, Mate1, and Pmat were not changed in the SA group. Taken together, repeated administration of RGE differentially regulated mRNA levels of Oct1 depending on the tissue types. However, mRNA levels of Oct2, Mate1, and Pmat were not changed in the 1WRA group compared with those in the control group. 

To investigate whether the single or repeated administration of RGE could change the protein expression of Oct1 and Oct2, we performed Western blot analysis of Oct1 and Oct2 transporters using the liver, kidney, and enterocyte tissue lysates of rats treated with RGE. As shown in Figure 3, Oct1 protein levels significantly increased in the enterocytes of rats in the 1WRA group compared to that in the control or SA groups. However, the Oct1 protein levels significantly decreased in the liver tissues of rats in the 1WRA group compared to that in the control or SA groups. Consistent with the mRNA results, Oct1 and Oct2 protein expression in the kidney was not affected by the single or multiple administration of RGE.

### 3.4. The Effect of RGE Treatment on the Intestinal Permeability of Metformin

To investigate whether the metformin absorption was affected by the treatment of RGE or not, we measured the intestinal permeability of metformin using the duodenal, jejunal, and ileal segments from rats of control, SA, and 1WRA groups. Absorptive permeability of metformin in the duodenum and ileum significantly increased in 1WRA group (Figure 4A). However, the metformin permeability in the jejunum tended to increase but did not reach statistical significance (*p* > 0.05) in 1WRA group. Consistent with mRNA expression levels of Oct1, metformin permeability in duodenum, jejunum, and ileum was not changed in SA group.

For the comparison, we also measured the permeability of 3-*O*-methyl-d-glucose, a marker compound of glucose transporters [32], in the duodenum, jejunum, and ileum. As results, the absorption of 3-*O*-methyl-d-glucose was much higher than that of metformin in all intestinal segments but not affected by the single or multiple treatment of RGE (Figure 4B). The permeability of Lucifer yellow, a marker compound for cell integrity and paracellular permeability [31,35], was not altered in rats of SA and 1WRA group (Figure 4C). The results suggested that the absorption of metformin from the duodenum and ileum was enhanced by the repeated administration of RGE without altering the absorption of glucose and cell integrity, which could consequently cause the increased absorption and plasma concentrations of metformin, including *C*_max_ and AUC value. 

### 3.5. Effect of RGE on the PK of metformin in rats

We investigated the effect of RGE on the metformin PK following the intravenous injection of metformin at a dose of 0.25 mg/kg and the results are shown in Figure 5A and Table 2.

Plasma concentrations of metformin were not affected by the single or 1-week repeated administration of RGE (1.5 g/kg/day) and the PK parameters from these three groups were not statistically different. The urinary recovery of metformin after the intravenous injection was about 80%, which was consistent with the previous report [4], and was not changed by the treatment of RGE (Table 2). The results suggested that neither single administration nor 1-week repeated administration of RGE treatment affected the elimination of metformin. 

Next, we measured the plasma concentrations and renal excretion of metformin following oral administration at a dose of 50 mg/kg after the single or 1-week repeated administration of RGE (1.5 g/kg/day) (Figure 5B). PK parameters of metformin calculated from the plasma concentrations and renal excretion of oral metformin are shown in Table 3. 

No statistical differences were observed in any PK parameters of metformin between control and SA groups. However, in the repeated RGE treatment group (1WRA), the elimination half-life (*T*_1/2_), *C*_max_, AUC, and fraction of the metformin dose excreted into urine as unchanged drug (Ae_24h_) were increased compared with those in control group (Table 3). The results suggest that multiple dose of RGE may increase the plasma concentration and renal excretion of metformin. Considering that intravenous metformin was not affected by the multiple treatment of RGE, the increased metformin concentrations and urinary excretion following its oral administration could be explained from the increased absorption of metformin, which is also consistent with the increased absorptive permeability of metformin and the increased intestinal Oct1 expression by the repeated RGE treatment (Figure 2, Figure 3 and Figure 4).

## 4. Discussion

In this pharmacokinetic herb–drug interaction study, a single dose of RGE was administered to investigate whether RGE treatment could directly affect the ADME (absorption, distribution, metabolism, and excretion) or PK properties of metformin through the inhibition of critical pathways, such as OCT and MATE. Similarly, multiple administration of RGE was performed to investigate whether RGE treatment could regulate the expression of drug transporters (Oct1, Oct2, Mate1, and Pmat) that affect the ADME or PK properties of metformin. Therefore, we investigated the inhibitory effects of RGE on in vitro OCT1, OCT2, MATE1, and MATE2-K transport activities and investigated the mRNA levels of Oct1, Oct2, Mate1, and Pmat in the duodenum, jejunum, and ileum, liver, and kidney from rats treated with multiple dose of RGE. Consequently, RGE was not found to inhibit OCT1 and OCT2 transport function; however, MATE1 and MATE2-K transport functions were inhibited at high concentrations of RGE. Its repeated administration (1.5 g/kg/day) for 1 week increased the Oct1 expression in the duodenum, jejunum, and ileum but decreased it in the liver; however, the expression of Oct1, Oct2, Mate1, and Pmat in the duodenum, jejunum, and kidney remained unchanged. 

According to the previous report, the inhibition of in vitro transport activities could be indicative of herb–drug interaction potential, along with the peak plasma concentration, protein binding, and IC_50_ values of interacting components [36]. That is, when the values calculated from the equation of peak plasma concentration/IC_50_ are below 0.1, the herb–drug interaction potential caused by direct inhibition would be expected to be remote [36]. However, RGE contains various ginsenosides, such as Rb1, Rb2, Rc, Rd, Rg1, Rg3, and Rh1, which get further metabolized into small molecular weight ginsenosides, such as F1, F2, and compound K [37]. The steady-state plasma concentrations of ginsenosides detected in rat plasma following single or repeated administration of RGE were in the range of 2.9–35.7 ng/mL (Table 1), and therefore, calculated peak plasma concentration/IC_50_ values was far below 0.1. This expectation was consistent with the unchanged pharmacokinetic interaction between RGE and metformin following single administration of RGE (Figure 5).

However, repeated administration of RGE changed the Oct1 expression in the liver and intestine in a reciprocal direction, which could affect the absorption and disposition of the metformin, respectively [38]. Therefore, in vivo PK study was considered to be necessary to determine whether the tissue differential regulation of Oct1 by RGE was relevant or not. First, metformin was intravenously administered to investigate whether herb–drug interaction between RGE and metformin could occur in the elimination process of metformin. PK profile and urinary excretion of metformin following intravenous injection were not changed in 1WRA groups or in SA group. Although the hepatic Oct1, which could contribute to the distribution of metformin from the plasma to the liver, was decreased by the multiple RGE treatment, it did not alter the PK of metformin. Since other metformin transporters, such as hepatic Mate1, renal Oc1, Oct2, and Mate1, were not changed in 1WRA group, they played minor roles in the herb–drug interaction between RGE and metformin.

Metformin is known to be broadly absorbed from the duodenum to ileum [5,39] with a moderate permeability of metformin (5.5 × 10^−6^ cm/s) in Caco-2 cells [40], which is a comparable value with our results (1.2–4 × 10^−6^ cm/s; Figure 4). The intestinal absorption of metformin could occur via transcellular mechanism involving an active and saturable process, as well as passive diffusion; however, the respective transporters could not be specified [40]. Another study reported that the intestinal absorption process is mediated by transporters, such as Oct1, Pmat, and Mate1 [10]. Metformin is mainly absorbed in the jejunum and ileum (60% of oral dose) [5] and the expression of Oct1 is the most abundant in the rat ileum [29]. Based on these reports, we deduced that the increased Oct1 in the rat intestine could contribute to the absorption of metformin, and consequently, increase the plasma concentration of this drug.

Therefore, we investigated PK interaction of oral metformin following single or repeated administration of RGE for 1 week to elucidate whether the enhanced Oct1 expression and metformin permeability could contribute to the PK interaction of metformin. As a result, plasma exposure (*C*_max_ and AUC) of metformin was increased and urinary excretion (Ae_24h_) of this drug was also increased, which could be explained from the increased absorption of metformin rather than the alteration of urinary excretion rate of metformin. T_1/2_ of metformin following oral administration (3.53 ± 0.74 h) was longer than that following intravenous injection (1.10 ± 0.25 h), suggesting the absorption of metformin could occur in the broad area of the intestine, consistent with previous reports [5]. In this study, the increased absorption of metformin could occur in the duodenum and ileum (Figure 4A), and thus it could lead to the apparent increase in *T*_1/2_ (Table 3). 

In rats, intestinal first-pass effect contributed to the low oral bioavailability of metformin and Cyp2c11, 2d1, and 3a1/2 were involved in the metabolism of metformin [39,41]. Therefore, the inhibitory effect of RGE on the metabolism of metformin could contribute to the increased absorption of metformin. Previously, Jo and Lee reported no PK interaction between single oral dose RGE (0.5–2.0 g/kg) and 5 Cytochrome P450 enzymes (i.e., Cyp1a, Cyp2B, Cyp2C, Cyp2d, and Cyp3a) [42]. They also reported repeated oral dose of RGE (0.5 g/kg for 2 weeks) did not alter the metabolic activity of Cyp1a, Cyp2b, Cyp2C, Cyp2d, and Cyp3a in the liver [43]. Another study reported that oral administration of ethanol extract of ginseng (30 mg/kg for 10 days, 8.13 mg/kg ginsenosides as contents) to rats increased the mRNA levels of Cyp2d1 and Cyp3a2 in the liver [44]. However, they did not measure the alterations in the metabolic activity or expression of intestinal metabolizing enzymes, which remain to be investigated to understand the mechanisms underlying the herb–drug interaction between RGE and oral metformin, as well as the increased absorption of this drug via the increased expression of Oct1 in the duodenum, jejunum, and ileum. 

Transcriptional regulation of murine Oct1 was regulated by the presence of hepatocyte nuclear factor-4α (Hnf-4α), Hnf-1α, and peroxisome proliferator agonist receptor (PPAR)-γ [34,45,46]. In addition, mRNA level of rat Oct1 and Oct2 was increased by the treatment of pregnane X receptor (PXR) ligand in the primary cultured cells and rat liver and kidney [47]. At present, transcription factors that are involved in the differential regulation of Oct1 by the repeated administration of RGE could not be specified. The expression of Oct1 was decreased in the intestine but not changed in the liver and kidney in Hnf-1α null mice [45]. The levels of HNF-1α were correlated with the expression of OCT1 in HepG2 and Huh7 cells [48]. The oral administration of ginseng extract (30 mg/kg for 10 days) to rats increased mRNA levels of Hnf-1α in the liver [44]. Since the RGE or ginsenoside concentrations in the intestinal lumen were higher than those in the liver or kidney, the differential regulation of Oct1, depending on the tissue type, could be explained by the differential activation of transcription factors, such as Hnf-1α.

RGE is the most popular product of ginseng (the roots and rhizomes of Panax ginseng C.A. Meyer) and showed anti-diabetic potential in animals and humans. Because of its anti-diabetic effect, we investigated herb–drug interaction potential between metformin and concomitantly administered RGE. The results showed that RGE moderately increased ileal Oct1 but decreased its hepatic expression without inhibiting the in vitro transport activity. This resulted in increased absorption of metformin from the lower part of the small intestine, consequently increased the in vivo plasma concentration of metformin, but delayed the elimination rate of metformin in rats. Since more than 40% of clinically used drugs are organic cations and cationic transporters are important for their PK and therapeutic effect [34], the herb–drug interaction potential between RGE and Oct1/OCT1 substrate drugs, including metformin, should be considered before establishing therapeutic regimens.

## Figures and Tables

**Figure 1 pharmaceutics-11-00189-f001:**
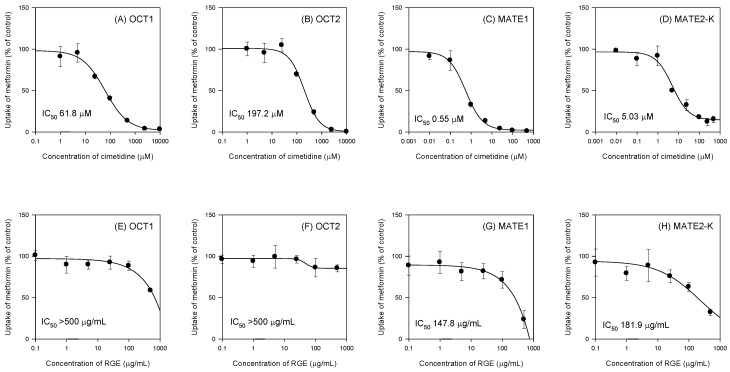
Inhibitory effect of cimetidine (**A**–**D**) and Korean red ginseng extract (RGE) (**E**–**H**) on organic cation transporter 1 (OCT1), OCT2, multiple toxin and extrusion protein 1 (MATE1), and MATE2-K-mediated uptake of metformin. Effect of cimetidine (0.01 µM–10 mM) and RGE (0.1–500 µg/mL) on uptake of 10 µM [^14^C]metformin in HEK293 cells overexpressing OCT1, OCT2, MATE1, and MATE2-K transporters was measured for 5 min. Data points represent means ± SD of three independent experiments. Data were fitted to an inhibitory effect Sigmoid Emax model and the IC_50_ value was calculated.

**Figure 2 pharmaceutics-11-00189-f002:**
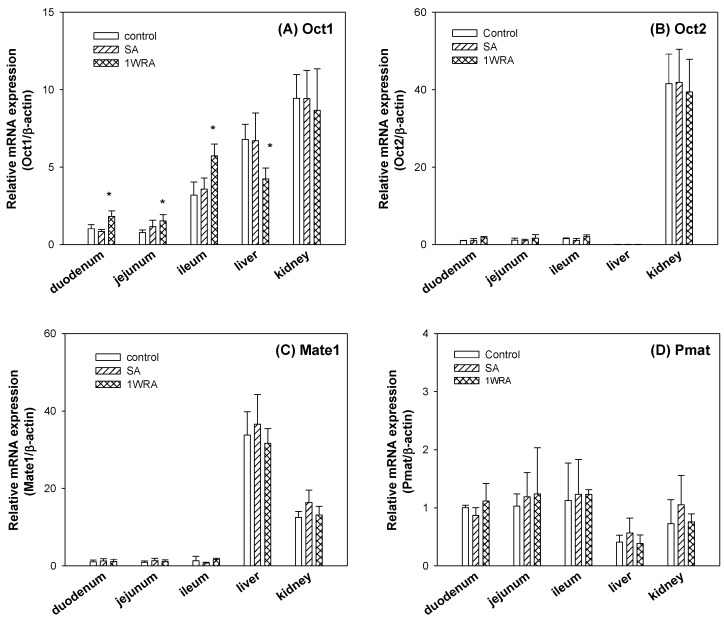
The mRNA expression of organic cation transporter 1 (Oct1) (**A**), Oct2 (**B**), multiple toxin and extrusion protein 1 (Mate1) (**C**), and plasma membrane monoamine transporter (Pmat) (**D**) in the duodenum, jejunum, ileum, liver, and kidney of rats in control, single RGE treatment (SA), and 1-week repeated RGE administration (1WRA) groups. Bars represent mean ± SD of three independent analyses; * *p* < 0.05 compared with control group (by Student’s *t*-test).

**Figure 3 pharmaceutics-11-00189-f003:**
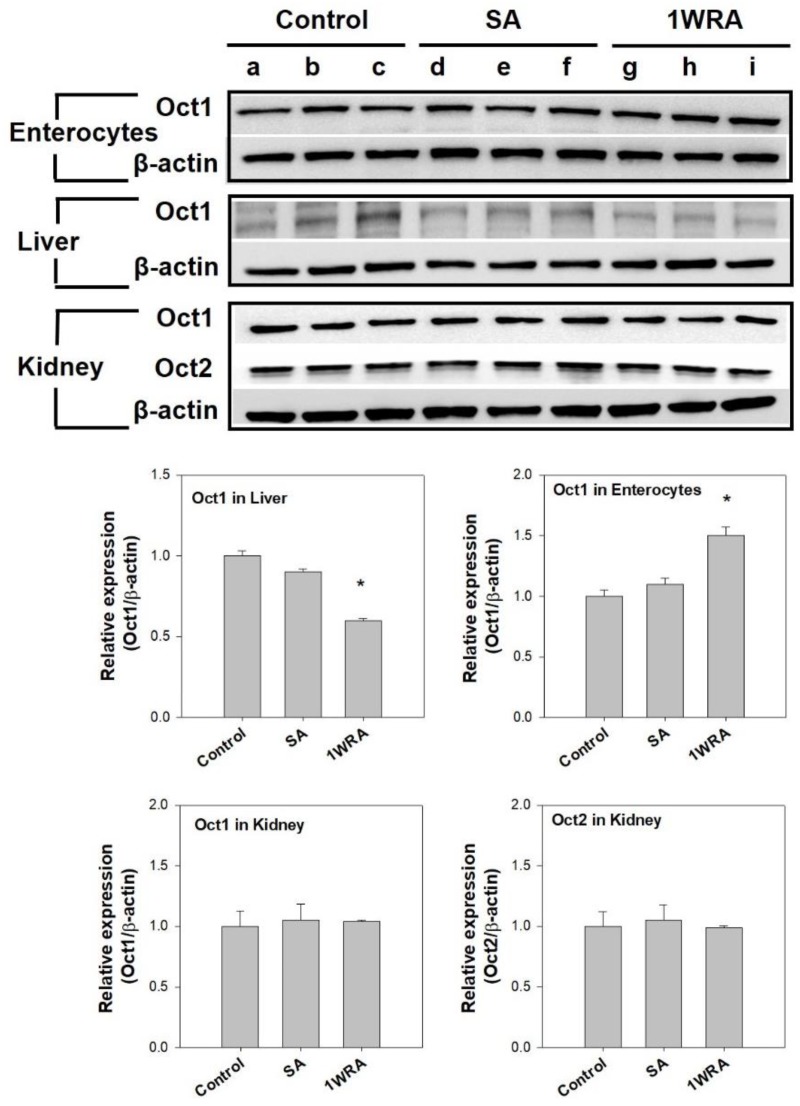
Protein expression of organic cation transporter 1 (Oct1) in the ileal enterocytes, liver, and kidney and Oct2 in the kidney of rats in control (lanes a, b, and c), single RGE treatment (SA, lanes d, e, and f), and 1-week repeated RGE administration (1WRA, lanes g, h, and i) groups and β-actin was used as a loading control. Quantitative analysis of western blot results is shown in the lower panel. Bars are mean ± SD of three independent densitometric analyses; * *p* < 0.05 compared with control group (by Student’s *t*-test).

**Figure 4 pharmaceutics-11-00189-f004:**
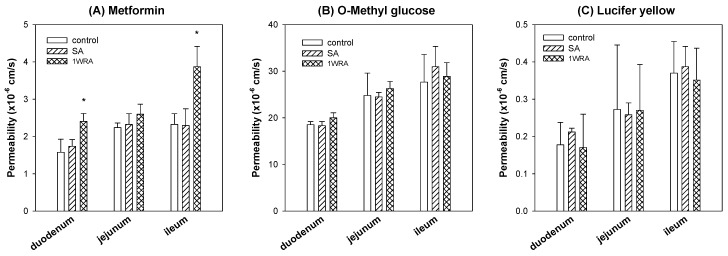
Intestinal absorptive permeability (P_app_) of metformin (**A**), 3-*O*-methyl-d-glucose (**B**), and Lucifer yellow (**C**) was measured in duodenum, jejunum, and ileum segments from rats of control, single RGE treatment (SA), and 1-week repeated RGE administration (1WRA) groups. Data points represent the mean ± SD from three different rats per group; * *p* < 0.05 compared with control group (by Student’s *t*-test).

**Figure 5 pharmaceutics-11-00189-f005:**
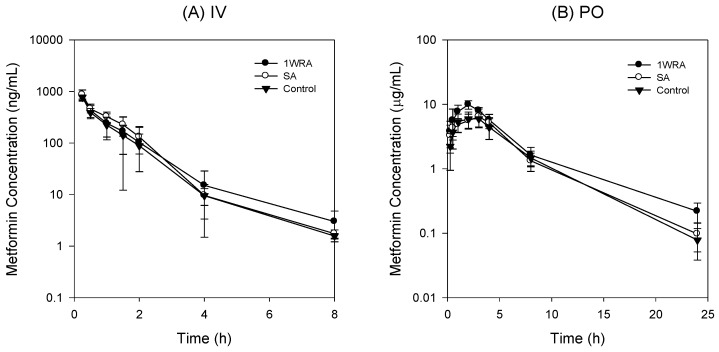
Plasma concentration-time profile of metformin in control, single administration of RGE (SA), and 1-week repeated RGE administration (1WRA, 1.5 g/kg/day) group following intravenous injection (IV) of metformin at a dose of 0.25 mg/kg (**A**) and oral administration (PO) of metformin at a dose of 50 mg/kg (**B**) Data points represent the mean ± SD from five different rats per group.

**Table 1 pharmaceutics-11-00189-t001:** Concentrations of ginsenosides in red ginseng extract (RGE) and rat plasma.

Ginsenosides		RGE	Rat Plasma	
		(μg/g RGE)	SA (ng/mL)	1WRA (ng/mL)
20(s)-protopanaxadiol	Rb1	2169.5 ± 312.1	14.3 ± 2.7	35.7 ± 1.9
	Rb2	856.3 ± 104.4	6.3 ± 2.1	29.4 ± 1.5
	Rc	1385.8 ± 164.0	9.0 ± 1.6	27.1 ± 1.3
	Rd	307.7 ± 48.4	2.9 ± 0.6	14.4 ± 2.6
	Rh2	ND	ND	ND
	Rg3	1534.9 ± 109.0	ND	ND
	F2	ND	ND	ND
	Compound K	ND	ND	ND
	Protopanaxadiol	ND	ND	ND
20(s)-protopanaxatriol	Re	528.3 ± 78.9	ND	ND
	Rh1	490.7 ± 90.1	ND	ND
	Rg1	999.4 ± 43.2	ND	ND
	F1	ND	ND	ND
	Protopanaxatriol	ND	ND	ND

Data are expressed as mean ± SD from six independent measurements of ginsenosides in RGE and rat plasma 2 h after the single administration of RGE (1.5 g/kg) or 24 h after the last RGE administration (1.5 g/kg/day). RGE: Red ginseng extract. SA: single administration of RGE. 1WRA: repeated administration of RGE for 1 week. ND: Not detected.

**Table 2 pharmaceutics-11-00189-t002:** Pharmacokinetic parameters of metformin in control, single oral administration (SA), and 1-week repeated administration (1WRA) of RGE (1.5 g/kg/day) following intravenous injection of metformin at a dose of 0.25 mg/kg in rats.

PK Parameters	Control	SA	1WRA
*T*_1/2_ (h)	1.10 ± 0.25	0.97 ± 0.16	1.32 ± 0.34
*C*_0_ (ng/mL)	1565.61 ± 247.60	1672.64 ± 623.06	1757.40 ± 437.84
AUC_24h_ (ng·h/mL)	830.37 ± 211.80	1047.10 ± 236.94	980.12 ± 361.96
AUC_∞_ (ng·h/mL)	840.74 ± 212.54	1054.51 ± 239.22	985.62 ± 363.63
MRT (h)	0.70 ± 0.23	0.81 ± 0.17	0.85 ± 0.35
Ae_24h_ (%)	80.07 ± 4.93	83.52 ± 5.72	79.56 ± 5.02

Data represent mean ± SD of five rats per group. *T*_1/2_: elimination half-life; *C*_0_: initial plasma concentration; AUC_24h_ or AUC_∞_: Area under plasma concentration-time curve from zero to 24 h or infinity; MRT: mean residence time; Ae_24h_: the fraction of the dose excreted in urine for 24 h as a parent form.

**Table 3 pharmaceutics-11-00189-t003:** Pharmacokinetic parameters of metformin in control, single oral administration (SA), and 1-week repeated administration (1WRA) of RGE (1.5 g/kg/day) following oral administration of metformin at a dose of 50 mg/kg in rats.

PK Parameters	Control	SA	1WRA
*T*_1/2_ (h)	3.53 ± 0.74	3.56 ± 0.47	4.43 ± 0.74 *
*C*_max_ (μg/mL)	6.31 ± 1.68	6.47 ± 2.00	9.90 ± 1.49 *
*T*_max_ (h)	2.40 ± 0.89	2.50 ± 1.12	2.00 ± 0.01
AUC_24h_ (μg·h/mL)	44.31 ± 8.31	45.20 ± 11.12	57.62 ± 4.99 *
AUC_∞_ (μg·h/mL)	44.75 ± 8.09	45.72 ± 11.18	59.04 ± 5.17 *
MRT (h)	4.85 ± 0.58	4.69 ± 0.33	4.64 ± 0.68
Ae_24h_ (%)	54.56 ± 3.26	53.04 ± 3.69	66.29 ± 10.57 *

Data represent mean ± SD of five rats per group; * *p* < 0.05 compared with control group (by Student’s *t*-test). *T*_1/2_: elimination half-life; *C*_max_: maximum plasma concentration; *T*_max_: time to reach *C*_max_ ; AUC_24h_ or AUC_∞_: Area under plasma concentration-time curve from zero to 24 h or infinity; MRT: mean residence time; Ae_24h_: the fraction of the dose excreted in urine for 24 h as a parent form.

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
