# Peer review of "Enhanced Intestinal Permeability and Plasma Concentration of Metformin in Rats by the Repeated Administration of Red Ginseng Extract"

_pharmaceutics, 2019, doi:10.3390/pharmaceutics11040189_

Round 1
Reviewer 1 Report
The authors addressed all the concerns raised and manuscript can be recommended for publication in the present form.
Reviewer 2 Report
The authors have adequately dealt with my comments. I have nothing to add.
This manuscript is a resubmission of an earlier submission. The following is a list of the peer review reports and author responses from that submission.
Round 1
Reviewer 1 Report
1 In this study, the RGE contains various ginsenosides such as Rb1, Rb2, Rc, Rd, Rg1, Rg3, and Rh1. The authors should analyze these ginsenosides and present the concentrations of them.
2 How did the authors select the doses of RGE? Please introduce or discuss this issue.
3 In the introduction, the authors said “Based on the beneficial effect of RGE, metformin and RGE could be combined easily as metformin add-on therapy.” We want to know the truth of combination therapy. Is there some reports of herb-drug interaction in the clinic?
4 Why did the authors select one week (7 days) treatment? How about 14 days or one month?
5 Except of OCT, other transporters and drug metabolic enzymes are involved in the ADME of metformin. The authors should carry out more studies to make sure the results.
Reviewer 2 Report
Jin and coworkers investigated the impact of red ginseng extract on transporter expression and pharmacokinetic parameters of metformin in rats. To this end, the authors treated rats with red ginseng extract for up to a week and determined the expression of OCT1 and OCT2 in various tissues and performed a pharmacokinetic analysis of metformin. These experiments were complemented by inhibition studies of the transporters in transfected cell lines and by Ussing-chamber experiments with intestinal specimens from rats. The authors found no interaction of rat ginseng extract with OCT1 and OCT2 transport activity. OCT1 expression was increased in the intestine and decreased in the liver, while expression of OCT1 and OCT2 were not altered by the treatment in kidney. The pharmacokinetic analysis revealed an increased urinary excretion of metformin in treated rats and an increased plasma half-life.
Comments
1. English: There are a few minor glimpses.
2. Abstract: The authors state that the elimination half-life in the repeated treatment group was decreased, however the data in table 1 show an increase of the half-life. I suggest that the species used for the study is mentioned early in the abstract and that the route of administration is also given in the abstract.
3. Methods: How were the animals sacrificed? The authors mention enterocytes: How did they isolate the enterocytes from the rat intestines? What was the rationale to only analyze the ileum? Drug absorption typically occurs along the entire intestine. How was the ileum defined? Ussing chamber experiments: What quality control measurements were performed? mRNA and protein normalization in the intestine would be better performed against villin, which is only expressed in enterocytes. How was the tissue homogenized for Western blotting in RIPA buffer? What assay was used for protein determination?
4. Figure 4: The authors need to use a control, e.g. 3-O-methyl glucose or an amino acid to test, whether the permeability change is specific for metformin.
5. Discussion: What potential mechanism leads to a down-regulation of OCT1 in liver and to an upregulation of this transporter in the intestine? This should be discussed.
Reviewer 3 Report
Review of manuscript „Enhanced intestinal permeability and plasma concentration of metformin by the repeated administration of red ginseng extract”.
The manuscript describes evaluation of potential herb-drug interactions between Korean red ginseng extract and metformin in terms of modulation of organic cation transporters. Although the topic of the paper appears to be interesting, the authors did several mistakes which are listed below.
· The authors should be more careful English language; and typographical errors, eg. In abstract you have OCT and Oct, then: However, repeated administration of RGE (1.5 kg/kg/day for 1 week) differentially regulated the expression of Oct1 and Oct2 depends on the tissue type – this sentence is not correct. Introduction: For this, metformin add-on therapy with other drugs that show different mode of action has been recommend – should be recommended.
· The authors should provide justification for the dose of RGE.
· In abstract there is no information on the type of studies – in vivo? In what animal model?
· In introduction the authors should not mmmix the data on metformin with RGE, please write separate section, first on metformin, then RGE.
· The authors should focus not only on metformin elimination and role of OTCs, but also on absorption.
· In materials and methods section please provide the number of animals. Why only male rats were chosen for the studies?
· Validation of the methods should be performed.
· Figure 3 is not clear – what is % of control? Then the values on Y axis are 1.1 etc…
· The effect of RGE treatment on the intestinal permeability of metformin – can we assign the results with the direct effect of RGE on intestine cells?
· I suggest also to conduct the studies with radiolabeled metformin in HEK cells, then we could directly assess the effects of RGE in the metformin uptake.